# Fibroblasts Collagen Production and Histological Alterations in Hereditary Gingival Fibromatosis

**DOI:** 10.3390/diseases7020039

**Published:** 2019-05-25

**Authors:** Lourdes Roman-Malo, Beatriz Bullon, Manuel de Miguel, Pedro Bullon

**Affiliations:** 1Department of Periodontology, Facultad de Odontologia Universidad de Sevilla, c/Avicena s/n, 41009 Sevilla, Spain; lvrmalo@gmail.com (L.R.-M.); beatrizbullon@hotmail.com (B.B.); 2Department of Cytology and Normal Histology and Pathologogy, Facultad de Medicina, Universidad de Sevilla, Avda. Sánchez Pizjuán s/n, 41009 Sevilla, Spain; mmiguel@us.es

**Keywords:** gingival fibromatosis, gingival overgrowth, fibroblasts, collagen, oxidative stress, antioxidants

## Abstract

Hereditary gingival fibromatosis is a disorder for which the etiology remains unknown. We aimed to evaluate the fibroblasts and histological alterations to give new clues. A father and a daughter of a family showing gingival hereditary fibromatosis were treated, and gingival biopsies were obtained. A histological study revealed dense fibrous tissue, basal lamina disruption, and epithelial cell migration into the connective tissue. Fibroblasts were cultured from the father and daughter and compared with those from a healthy control patient. The results of the biochemical analysis showed increased collagen synthesis, reduced antioxidant CoQ_10_ content, and high levels of lipid peroxidation. Additionally, fibroblasts culture incubation with the oxidant H_2_O_2_ increased collagen levels that have been reduced by the addition of the antioxidant CoQ_10_. We conclude that some fibroblasts metabolic alterations play a significant role in initiating and maintaining persistent fibrotic tissue. Oxidative stress influences the fibroblasts collagen production and could play a particular role in the pathogenesis of hereditary gingival fibromatosis.

## 1. Introduction

Hereditary gingival fibromatosis (HGF), also called elephantiasis gingivae, hereditary gingival hyperplasia, and hypertrophic gingiva, is a disorder characterized by progressive enlargement of the gingiva. This enlargement results from an increase in the connective tissue elements of the submucosa and displays different severities, sometimes covering the entire crowns of the teeth and deforming the palate, thereby creating occlusal and aesthetic problems, as well as causing difficulties in speech and mastication. Thickening of the alveolar ridge rarely appears at birth, typically initiates with the eruption of deciduous or permanent dentition, exacerbates during adolescence, and can persist within adulthood [1]. The quantity of hyperplastic tissue varies among subjects, with coverage ranging from a small par, through to the whole crown of the tooth. In some cases, the growth is more pronounced in the tuberosity area, causing deformity of the palate and impairing phonation and deglutition, even reaching the midline [2]. However radiographic imaging shows no specific changes in the teeth or alveolar bone.

The classification of HGF includes two types according to its form; the enlargement may be generalized, involving the entire gingivae of the maxilla and the mandible, or localized to particular areas of the mouth, such as the maxillary tuberosities and the labial gingiva around the lower molars. HGF may also exhibit an autosomal dominant or recessive mode of inheritance. The recessive pattern usually linked to other syndromes: Cowden, Jones, Goltz-Gorlin, Murray-Puretic-Drescher, Ramon, Rutherfurd, Cross; and other systemic diseases: cherubism, hypothyroidism, chondrodystrophy, growth hormone deficiency, hypertrichosis, epilepsy, mental retardation, craniofacial dysmorphism or leukemia [3,4,5,6]. Also, gingival overgrowth has been associated with ingestion of drugs such as phenytoin, nifedipine, verapamil, and cyclosporine [7]. It is important in such cases to establish a differential diagnosis in order to prevent unnecessary interventions and create an adequate treatment plan according to the etiology.

Oral manifestations of HGF include particularly thick and fibrous gingiva with relatively little inflammatory change. The enlarged tissues are firm, pink and nodular in appearance and may show exaggerated stippling. These tissues are non-erythematous, except for locally caused inflammation, and have no tendency to bleed.

Histologically, HGF shows gingival features relatively acellular and with an increased amount of randomly arranged bundles of collagen. The overlying epithelium may be variable in thickness and have prominent, elongated rete ridges extending into the underlying connective tissue [8]. The typical histologic appearance of the gingival lesions in familial gingival fibromatosis consists of hyperplastic dense fibrous connective tissues that are formed by thick bundles of collagen fibers, small calcified particles, islands of osseous metaplasia, ulceration of the overlying mucosa, and inflammation [9]. In rare cases, the description includes deposition of amyloids and islands of odontogenic epithelium [10]. However, these histological features are nonspecific, and the diagnosis should be based on clinical findings and family history [11].

Although the etiology of this pathology is unknown, several theories have been proposed to explain its development [6]. The focus of research has been on factors related to fibrosis including cellular proliferation, collagen synthesis, extracellular matrix degeneration, cytokines or growth factors. These conditions are presented with the epithelial to mesenchymal transition (EMT), where the basal lamina show disruptions and epithelial cells migrate into connective tissue and change their phenotypes to fibroblast-like cells [12]. Fibroblasts are the key cells involved in the gingival production of collagen and respond to the local stress depending on the environmental conditions. Therefore, is essential that they have to maintain a good metabolic statement to respond to all aggressions. One of the main expressions of the metabolic statement is oxidative stress; it is the imbalance between reactive oxygen species (ROS) and antioxidant mechanisms. More ROS increase the normal redox state of cells and damage all components of the cell, including proteins, lipids, and DNA [13]. Chronic periodontitis is related to oxidative stress [14], and previous studies have found a relationship between oxidative stress and cyclosporine-induced gingival overgrowth [15], but to date, no studies have linked this oxidative stress to HGF.

We describe the histological features of a family with HGF and analyze in fibroblasts culture collagen production that can help to understand the pathogenic process.

## 2. Case Report

A 50-year-old male presented at the Dental School, University of Seville, with a chief complaint of “gingival swelling all over his mouth”. After the completion of the informed consent, the cases were studied. The swelling caused difficulties in speaking and chewing. No other complaints including pain, bleeding, or halitosis were present. The patient began 4–5 years ago and progressed slowly. His medical history was unremarkable, and he was not taking any medication. His weight and height were within normal limits. An extraoral examination revealed a natural face. The patient could close his lips; however, he was an oral breather. An intraoral examination presented signs of gingival enlargement, a bilateral fullness of the cheeks with painless swelling and palate extension to the midline (Figure 1A–D). We suggested an examination of his family. His wife suffered chronic periodontitis without any gingival enlargement. He had two daughters: one ten years old and the other twelve years old. The younger daughter presented with regular aspects of the gums and the oldest daughter presented with generalized enlargement similar to her father (Figure 2A–D). According to these data, a diagnosis of hereditary gingival fibromatosis was established. The treatment consisted of a rigorous oral hygiene program in addition to professional debridement and gingivectomy at both maxillary quadrants.

## 3. Methods and Results

### 3.1. Histology

Gingival biopsies were obtained from the gingivectomy (Figure 3A). Hematoxylin-eosin staining and immunohistochemistry to detect laminin 5 (Menarini Diagnostics^®^), a protein present in an intact basement membrane, were performed using the automatic method (Ventana^®^) with a pre-diluted antibody in a Benchmark XT stainer (Roche Diagnostics). The histopathologic appearance of the gingivectomy specimens showed elongated papillae (Figure 3B), dense fibrous tissue, with few cells and mild inflammatory cell infiltrates (Figure 3C). We found some characteristics of EMT, including basal lamina disruption and epithelial cell migration into connective tissue (Figure 3D). The quantity of laminin, in both patients, presented a lack of laminin 5 in their basal membrane, which was more pronounced in the father (Figure 3E,F).

### 3.2. Fibroblast Culture

To evaluate the fibroblasts collagen production, we obtained gingival fibroblasts. Healthy control and patient gingival fibroblasts were cultured in Dulbecco’s Modified Eagle’s medium (DMEM) media (4500 mg/L glucose, L-glutamine, pyruvate, Gibco, Invitrogen, Eugene, Oregon, USA) supplemented with 10% fetal bovine serum (Gibco, Invitrogen, Eugene, Oregon, USA) and antibiotics (Sigma Chemical Co., St Louis, MO, USA). The cells were incubated at 37 °C in a 5% CO_2_ atmosphere and divided among three plates. The first plate did not receive any treatment; the second was treated with 100 µM H_2_O_2_, and the third was treated with 100 µM H_2_O_2_ and 30 µM CoQ_10_. After 48 h, pellets were obtained from each plate, and the collagen production was assayed in duplicate using a commercial kit according to the description of the protocol by the manufacturers (Sircol Collagen Assay, Biocolor Ltd., Belfast, UK).

### 3.3. Collagen Assay

The collagen levels in gingival tissue were determined by a method based on Sirius Red staining using a commercial kit (Sircol Collagen Assay, Biocolor Ltd., Belfast, UK). In these assays, a collagen standard was used to construct a standard curve against which unknown samples can be plotted. We observed increased collagen synthesis in both patients, with higher levels obtained with culture cells from the father (Figure 4A).

### 3.4. CoQ_10_ Level Determination

Gingival fibroblasts were lysed with 1% SDS and vortexed for 1 min. A mixture of ethanol: isopropanol (95:5) was added, and the samples were vortexed for 1 min. To recover CoQ, 5 mL of hexane was added, and the samples were centrifuged at 1000× *g* (5 min at 4 °C). The upper phases from three extractions were recovered and dried using a rotary evaporator. Lipid extracts were dispersed in 1 mL of ethanol, dried in a speed-vac and stored at –20 °C. Samples were suspended in a suitable volume of ethanol before high-performance liquid chromatography (HPLC) injection. The lipid components were separated by a Beckmann 166–126 HPLC system equipped with a 15-cm Kromasil C-18 column in a column oven set to 40 °C, with a flow rate of 1 mL/min and a mobile phase containing 65:35 methanol/n-propanol and 1.42 mM lithium perchlorate. CoQ levels were analyzed with ultraviolet- (System Gold 168), electrochemical- (Coulochem III ESA) or radioactivity- (Radioflow Detector LB 509, Berthold Technologies, Bad Wildbad, Germany) based detectors as necessary. Coenzyme Q_9_ (CoQ_9_) was used as the internal standard.

The levels of CoQ_10_, a mitochondrial cofactor with the potential to enhance mitochondrial function and antioxidant effects, were determined in gingival fibroblasts isolated from the patients and were lower in the father compared to both healthy control and daughter (Figure 4B). For the examination of the effects of oxidative stress on collagen synthesis, HGF cells were incubated with 100 µM H_2_O_2_ for 48 h. Collagen synthesis was significantly increased with upon H_2_O_2_ exposure, with levels restored through the addition of the antioxidant CoQ_10_ (Figure 4C).

### 3.5. Lipid Peroxidation

Thiobarbituric acid reactive substance (TBARS) levels within gingival fibroblasts were determined by a method based on a reaction with thiobarbituric acid (TBA) at 90–100 °C using a commercial kit from Cayman Chemical Company (Ann Arbor, MI). TBARS levels are expressed as malondialdehyde (MDA) levels. In these assays, an MDA standard was used to construct a standard curve against which unknown samples were plotted. Lipid peroxidation could be a consequence of basal ROS overproduction or fewer antioxidants capacity and could indicate high levels of oxidative stress in gingival fibroblasts from the father compared to the control and his daughter (Figure 4D).

### 3.6. Statistical Analysis

All results are expressed as the means ± SD unless stated otherwise. A one way ANOVA test was used to evaluate the significance of differences between the control and the patients, with *p* < 0.05 defined as the level of significance.

## 4. Discussion

This paper reports two cases of HGF. Generalized gingival enlargement may be the result of various causes, including inflammation, systemic diseases, drugs, or pregnancy. None of those conditions fit our patient’s case. Another form of gingival enlargement appears early in childhood and is known as idiopathic hyperplasia or HGF. This clinical diagnosis was confirmed by detection of hyperplastic dense fibrous connective tissue after biopsy of the maxilla.

Gingival enlargement typically begins at the time of eruption of the permanent dentition [16] or less frequently, with the eruption of the primary dentition [17]. In this case, the oldest patient noted only 4–5 years of development. Because such a generalized lesion typically requires more than 4–5 years to develop, we suspect that this period began with the onset of more severe limitations of speaking and eating.

The fact that only one of his two daughters presented with natural gums and the other daughter had general enlargement similar to her father suggests autosomal recessive inheritance. Previously, when idiopathic gingival fibromatosis was not associated with any syndrome, it has been described as autosomal dominant inheritance [18]. We believe that our patients are unique cases and reported as recessive.

The histologic features observed in our patients had the typical appearance of the gingival lesions of HGF including hyperplastic dense fibrous connective tissue with acanthotic gingival epithelium and elongated rete pegs [19]. We also found some features that have previously been described for the EMT [20]. The connective tissue contained cells with a fibroblast-like phenotype, and the laminin level indicated a loss of basement membrane integrity in both patients. About laminin 5, similar data have been found for drug-induced gingival overgrowth [21] but not in HGF.

Our main results from the fibroblasts culture showed that fibroblasts produce more collagen and present low level of CoQ_10_, and the addition of an oxidant such as H_2_O_2_ increase the collagen production that was restored by CoQ_10_.

Researchers agree that the connective tissue presents a high quantity of collagen in HGF. The controversial issue is why collagen is increased. According to one theory, greater accumulation of the extracellular matrix is a result of increased collagen synthesis and lack of degradation [22,23]. Our data show an increase in the synthesis of collagen but we have not studied collagen degradation produced by metalloproteinases as has been made previously [24].

Our results showed that the daughter has less degree of gingival overgrowth as we compare with the father. These are in accompanied by fewer collagens, a decrease in CoQ10 and also not an increase in MDA. These aspects can be due to the age differences.

Oxidative stress is related to apoptosis and cellular death; but also new data from in vitro assays have suggested that oxidative stress can also regulate collagen metabolism in cardiac fibroblasts and uterosacral ligament fibroblasts [25,26]. This is the first time that oxidative stress is described as a possible factor in the etiology of HGF. Becerik et al. [15] observed changes in oxidative stress markers for cyclosporine-induced gingival overgrowth. We found increased levels of lipid peroxidation as well as lower CoQ_10_ levels within fibroblasts that could indicate oxidative stress.

## 5. Conclusions

The histologic data showed basal lamina disruption, epithelial cell migration into connective tissue and a lack of laminin 5 in their basal membrane. In vitro results have demonstrated, for the first time, that collagen synthesis is influenced by an oxidant and can be restored by an antioxidant in HGF fibroblasts. Further studies are needed to verify this relationship.

## Figures and Tables

**Figure 1 diseases-07-00039-f001:**
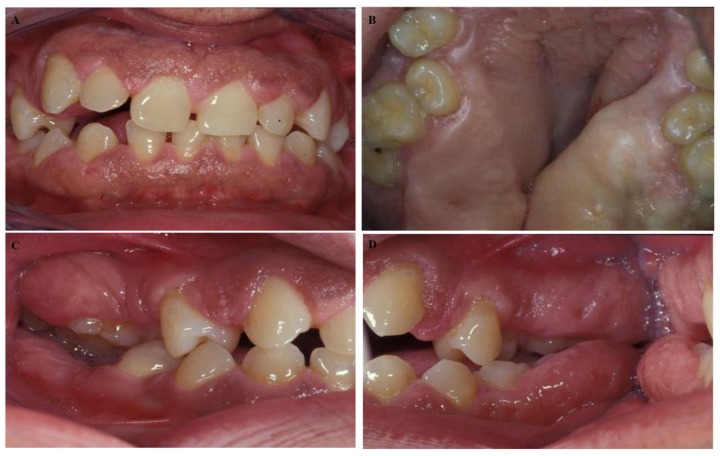
Intraoral images of a 50-year-old male. We observed (**A**) malocclusion with an anterior open bite, (**B**) rotation, tooth migration, (**C**,**D**) diastemas, malposition and severe gingival enlargement that caused difficulties in speech, mastication, and hygiene.

**Figure 2 diseases-07-00039-f002:**
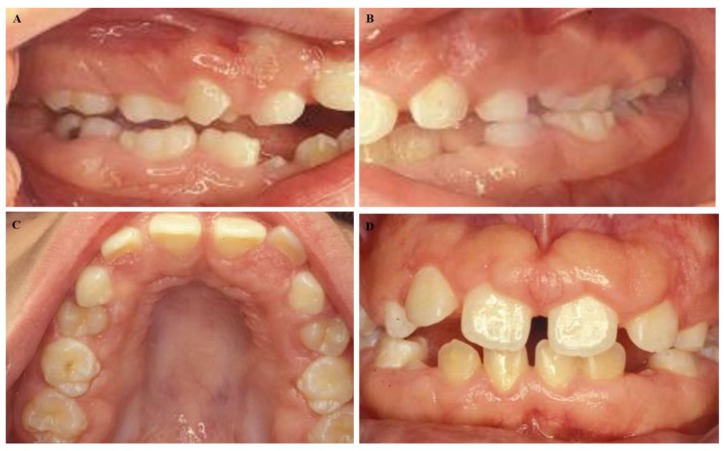
Intraoral images of the 12-year-old female daughter of the male patient in Figure 1. (**A**–**C**) We observed mixed dentition, (**D**) with malocclusion similar to her father, including an anterior open bite and delayed eruption.

**Figure 3 diseases-07-00039-f003:**
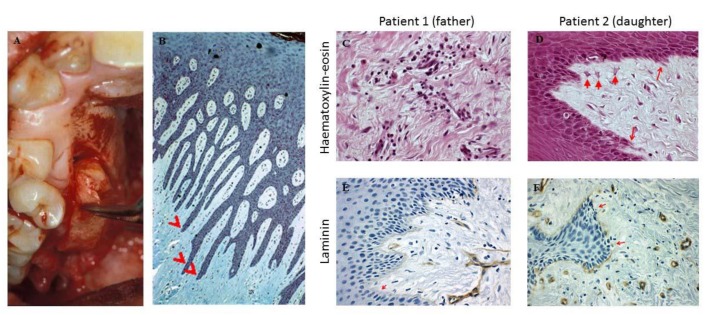
(**A**) Gingival specimens were obtained from the gingivectomy. (**B**) Histology of biopsies of both patients showed elongated papillae. (**C**) The inflammatory infiltrate and collagen fibers are shown (HE, 400x). (**D**) Arrows depict epithelial cell migration into connective tissue (HE, 400x). (**E**,**F**) The lack of laminin in the basal membrane is denoted by arrows (immune-histochemistry, laminin, 400×).

**Figure 4 diseases-07-00039-f004:**
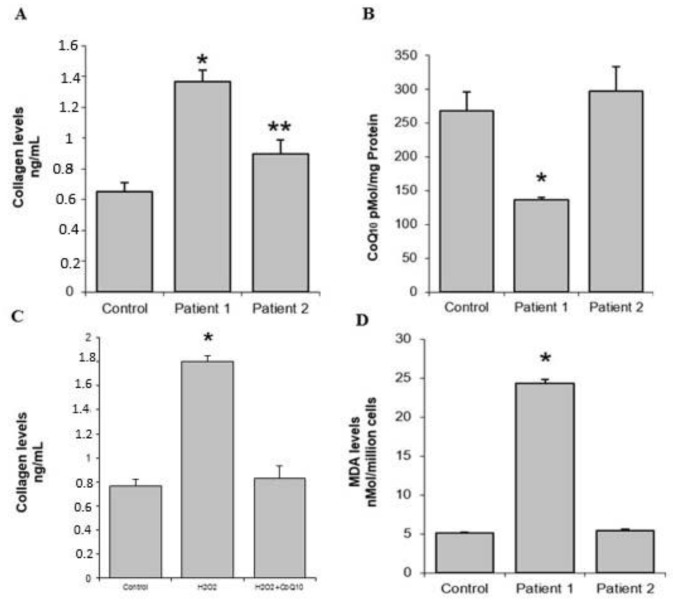
Biochemical determinations. (**A**) Collagen levels in gingival tissue; higher values were observed in patient 1. (**B**) The CoQ_10_ level was significantly decreased in the patient 1. (**C**) Collagen synthesis in patient 1 was significantly increased with oxidative stress; these levels were restored with the addition of the antioxidant CoQ_10_. (**D**) Lipid peroxidation in patient 1 was significantly increased. * = *p* < 0.01, ** = *p* < 0.05.

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
