# Peer review of "Fibroblasts Collagen Production and Histological Alterations in Hereditary Gingival Fibromatosis"

_diseases, 2019, doi:10.3390/diseases7020039_

Round 1
Reviewer 1 Report
The authors have described two cases of recessive gingival hereditary fibromatosis and have subsequently cultured fibroblasts from these patients and compared basal collagen content and basal antioxidant CoQ10 levels with a healthy control. They have then subjected HGF fibroblasts to oxidative stress induced by H2O2 (+/- CoQ10 supplementation) and also assayed lipid peroxidation within the cells.
Comments/Suggestions:
Abstract: Define CoQ10 and H2O2 (with the "2s" needing to be subscripted)
Line 21-22: needs to be re-written to make clearer, something along the lines of: "...incubation with H2O2 concurrently increased both collagen and lipid peroxidation, whilst supplementation with CoQ10 alleviated these effects"
Line 22-25: re-write: "...persistant fibrotic tissue, playing a particular role in the pathogenesis of HGF, with antioxidant supplementation a potential new therapeutic approach for the treatment of these patients."
Introduction: minor grammar changes needed, for example:
Line 32: change "show" to "displays"
Line 33: insert comma after "issues" change "and" to "as well as"
Line 36: change "in" to "within"
Line 37: after "part" insert ", through"
Line 39: Insert ", however radiographic" after Ref [2]
Line 41: insert "of HGF" after "classification"
Line 43: change "It may also" to "HGF may also"
Line 44: insert ", with the" after "inheritance"
Line 45: remove "appears"
Line 46: remove both "or"
Line 48: after [3-6], remove comma and insert "and is also"
Line 49: chose "those" to "such", change "the" to "a"
Line 50: insert "in order" following "diagnosis"
Line 60: change "that formed by" to "consisting of"
Line 63: Change "The" to "However these"
Line 66: Change "the reaction" to "its development" / chnage "Authors have focused their researchon" to "The focus of research has been on"
Line 76: insert "to date" following "but"
Line 78: "Here, we describe........synthesis from isolated patient gingival fibroblasts."
Insert heading "3. Methods and results" change Discussion heading to #4. Bold all subheadings (eg "Histology")
Line 105: Fig. 3E is the first decribed in text so Figure should be amended so that this image is Fig. 3A, so that results can be discussed in order. Also Fig. 3F is not mentioned anywhere in text and not descibed in Figure legend: please amend and describe.
Line 125: subscript the 2 in "CO2", also in H2O2 throughout.
Line 128: "previously described": Where? Need reference inserted, or change to "below" if this is the next method described.
Line 132: following "...both patients" insert: ", with higher levels obtained with cultured cells from the father (Patient 1).
Line 134-138: Figure legend should describe the methods used, not the results obtained.
Line 152: "skin fibroblasts" This is concerning, why were skin cells used here rather than the gingival used elsewhere? Also, form which patient were these obtained?
Line 155: Change "with oxidative stress..." to "upon H2O2 exposure, with levels restored through the addition of the antioxidant CoQ10 (Fig. 4C)."
Line 157: change "at" to "within"
Line 158-163: It is not clear whether or not this assay was done in resposne to H2O2 or measured under the basal conditions used in Fig. 4A-B, please rewrite to make clearer.
Line 164-166: Statistical analyis: The use of an unpaired student's t-test is not appropriate with three groups assayed, need to use one-way ANOVA (this also explains why the daughter is more significant (**) over the father (*) in Fig 4A, even though she has lower values overall. Please amend.
Line 181: fix Ref [16]
Line 201: The authors have not done any degradation studies ie looking at MMP and TIMP levels within the fibroblasts, this needs to be addressed and has been done previously, refer to work done:
J Periodontal Res. 2009 Dec;44(6):714-7. doi: 10.1111/j.1600-0765.2008.01180.x. Epub 2009 May 18.
Expression of metalloproteinases and their tissue inhibitors in gingiva affected by hereditary gingival fibromatosis: analysis of three cases within a family.
Gonçalves Lda R1, Oliveira GA, Borojevic R, Otazu IB, Feres-Filho EJ.
Line 207-208: Following "oxidative stress" insert " that corresponds to increased lipid peroxidation" as well as decreased antioxidant activity "corresponding to lower CoQ10 levels within fibroblasts"
Author Response
Comments Reviewer 1
Thank you for your valuable comments
The authors have described two cases of recessive gingival hereditary fibromatosis and have subsequently cultured fibroblasts from these patients and compared basal collagen content and basal antioxidant CoQ10 levels with a healthy control. They have then subjected HGF fibroblasts to oxidative stress induced by H2O2 (+/- CoQ10 supplementation) and also assayed lipid peroxidation within the cells.
Comments/Suggestions:
Abstract: Define CoQ10 and H2O2 (with the "2s" needing to be subscripted) changes have been made
Line 21-22: needs to be re-written to make clearer, something along the lines of: "...incubation with H2O2 concurrently increased both collagen and lipid peroxidation, whilst supplementation with CoQ10 alleviated these effects" ) changes have been made
Line 22-25: re-write: "...persistant fibrotic tissue, playing a particular role in the pathogenesis of HGF, with antioxidant supplementation a potential new therapeutic approach for the treatment of these patients." ) changes have been made
Introduction: minor grammar changes needed, for example:
Line 32: change "show" to "displays"
Line 33: insert comma after "issues" change "and" to "as well as"
Line 36: change "in" to "within"
Line 37: after "part" insert ", through"
Line 39: Insert ", however radiographic" after Ref [2]
Line 41: insert "of HGF" after "classification"
Line 43: change "It may also" to "HGF may also"
Line 44: insert ", with the" after "inheritance"
Line 45: remove "appears"
Line 46: remove both "or"
Line 48: after [3-6], remove comma and insert "and is also" 2
Line 49: chose "those" to "such", change "the" to "a"
Line 50: insert "in order" following "diagnosis"
Line 60: change "that formed by" to "consisting of"
Line 63: Change "The" to "However these"
Line 66: Change "the reaction" to "its development" / chnage "Authors have focused their researchon" to "The focus of research has been on"
Line 76: insert "to date" following "but"
Line 78: "Here, we describe........synthesis from isolated patient gingival fibroblasts."
Insert heading "3. Methods and results" change Discussion heading to #4. Bold all subheadings (eg "Histology")
Line 105: Fig. 3E is the first decribed in text so Figure should be amended so that this image is Fig. 3A, so that results can be discussed in order. Also Fig. 3F is not mentioned anywhere in text and not descibed in Figure legend: please amend and describe.
Line 125: subscript the 2 in "CO2", also in H2O2 throughout.
Line 128: "previously described": Where? Need reference inserted, or change to "below" if this is the next method described.
Line 132: following "...both patients" insert: ", with higher levels obtained with cultured cells from the father (Patient 1).
Line 134-138: Figure legend should describe the methods used, not the results obtained.
Line 152: "skin fibroblasts" This is concerning, why were skin cells used here rather than the gingival used elsewhere? Also, form which patient were these obtained?
Line 155: Change "with oxidative stress..." to "upon H2O2 exposure, with levels restored through the addition of the antioxidant CoQ10 (Fig. 4C)."
Line 157: change "at" to "within"
Line 158-163: It is not clear whether or not this assay was done in response to H2O2 or measured under the basal conditions used in Fig. 4A-B, please rewrite to make clearer. Lipid peroxidation is measured under basal conditions not in response to H2O2
Line 164-166: Statistical analyis: The use of an unpaired student's t-test is not appropriate with three groups assayed, need to use one-way ANOVA (this also explains why the daughter is more significant (**) over the father (*) in Fig 4A, even though she has lower values overall. Please amend.
Line 181: fix Ref [16] All these previous recommendations have been attended
Line 201: The authors have not done any degradation studies ie looking at MMP and TIMP levels within the fibroblasts, this needs to be addressed and has been done previously, refer to work done:
J Periodontal Res. 2009 Dec;44(6):714-7. doi: 10.1111/j.1600-0765.2008.01180.x. Epub 2009 May 18.
Expression of metalloproteinases and their tissue inhibitors in gingiva affected by hereditary gingival fibromatosis: analysis of three cases within a family.
Gonçalves Lda R1, Oliveira GA, Borojevic R, Otazu IB, Feres-Filho EJ.
We did not made any degradation technique just we demonstrated that more amount of collagen is produced, a comment has been added
Line 207-208: Following "oxidative stress" insert " that corresponds to increased lipid peroxidation" as well as decreased antioxidant activity "corresponding to lower CoQ10 levels within fibroblasts"
Changes have been made
Reviewer 2 Report
Lourdes Roman-Malo and coworkers present in this manuscript two related patients suffering from gingival overgrowth. The study describes the patients’ symptoms and investigates several parameters related to the disease in cultured fibroblasts of the patients. The authors find excessive collagen production and indications of oxidative stress in the patient cells.
While from the results presented here it becomes clear that the authors investigate an interesting and potentially central molecular aspect of the disease, this manuscript is combining the clinical aspects (patient symptoms and treatment) with the in vitro investigation on cell level. Unfortunately both parts suffer, as neither is properly presented and discussed.
The manuscript text is often imprecise (although the general meaning becomes clear), and the statements exaggerated. A thorough checking of every sentence for its actual appropriateness would be recommendable, and also the clear separation of actual results and hypothetical interpretation.
I recommend to either concentrate on the description of patient pathology, also elaborating on the treatment details and how it affected the development of the disease – or on the molecular reasons for familial hereditary gingivitis, leaving only a brief description of the patients, but conducting a range of more experiments to address the molecular processes.
My suggestion for a minimal set of experiments for the latter:
- quantification of ROS production using any of the usual fluorescent measurements in wt and patient1/2 fibros
- quantification of mitochondrial activity to confirm the low CoQ10 levels (e.g. MTA test)
- quantification of oxidative defenses (e.g. SOD2 or Glutathione)
Major critics:
- The description of the various types of HGF in the introduction is a bit confuse – after reading it I don’t understand which types exist and how they differ.
- CoQ10 is not an antioxidant used by the cell to defend against oxidative stress, but a part of the mitochondrial electron transport chain. It can be of course added as an antioxidant, but the quantification of endogenous CoQ10 does tell about the respiratory capacity, not about the antioxidant defense system of the cell. Please discuss (and preferably address experimentally) the mitochondrial etiology of the ROS stress and collagen production. Gingival overgrowth is a typical adverse effect of cyclosporin, that is known to screw up mitochondrial Calcium handling, again suggesting an alteration of mitochondrial function as the cause of the disease
- The patient’s treatment is elaborated in the discussion, but has actually not been the topic of the manuscript results.
- the fact that the patient’s fibroblasts produce excessive collagen also after culturing is not really discussed. As the cells are cultured in high glucose DMEM they are less dependent on mitochondrial oxidation, but still they suffer from oxidative stress and produce excessive collagen. That’s highly interesting and should not be ignored!
- The daughter has symptoms and collagen levels are elevated in her fibros, but these cells do not exhibit a decrease in CoQ10 and also not any increase in MDA. Thus the oxidative stress appears to kick in only at later stages of the disease and cannot be the cause of it – unless the cultured cells do not represent well the state in the tissue.
Please discuss this thoroughly, as the central message (oxidative stress as a cause of HGF) is very much questioned by this patient’s results!
- Whether the inheritance is recessive cannot be clearly judged by the absence of the symptoms in a 10 or 12 year old child, if the disease starts usually with the first permanent teeth – potentially it starts to show a bit later than usual. The father noticed problems only at age 45, so potentially there is some variations in the onset and progression.
small things:
line 2: here “gingival hereditary fibromatosis”, later on always “hereditary gingival fibromatosis”
line 3: in vitro and in vivo italic and without dash – all over the manuscript
line 30: should “natural” be rather “unnatural”?
line 32: show -> shows
line 33: issues -> problems, difficulties or similar
paragraph 29-40: this is somehow repetitive, twice mentioning the various degrees of overgrowth and the difficulties in speaking
line 45f: unclear whether Cowden, Jones, Goltz-Gorlin etc. is ONE syndrome or many syndromes? The whole sentence is a very long list that would be nicer spread over several sentences
line 48: “Also is associated”, something missing here, how about: “HGF is also associated”. In the same sentence “ingestion of drugs” probably means
line 49: the differential diagnosis -> a differential diagnosis
line 52: oral manifestations of the idiopathic type – which idiopathic type? Do you refer here to the autosomal dominant one? And are not all manifestations oral?
line 55: “inflammation, “ comma needed
paragraph 56ff: unclear to me whether the described lesions are belonging to the idiopathic type – if so, the topic suddenly jumps to familial cases in the next sentence – are they also idiopathic? If they refer to some other type please clarify which.
line 60: “that is formed”
line 68: this condition is related to a mechanism- how is it related?
line 70: the cell phenotype -> their phenotype
line 71: “and respond to the local stress depending on the metabolic conditions” – what exactly would you like to say here? Which local stress? Which metabolic conditions? How do the fibros react?
line 72: oxidative stress does not reflect the balance of ROS and antioxidants, but is an imbalance of these two
line 73: disturbances in the normal redox state could be also reduced ROS that are absolutely not harming any molecules, but only mess up the signaling
line 78: “we try to” – I assume you have succeeded with your try?!
line 83: does the informed consent needs to be mentioned here? it’s one case and not several cases
line 84f: no other complaints such as plain, bleeding or halitosis were present
line 85: He -> The patient… begun 4-5 years ago and progressed slowly
line 86: and he did not take any medication
line 87: within normal limits
line 99: severe enlargement of what?
line 105: “Histology” is a bit lonely – is it supposed to be a headline?
line 108: order information of the antibody missing
line 110: the epithelium was wider than what?
line 111: some characteristics of EMT
line 112ff: a bit weirdly phrased: the patients had the lack of laminin 5 all the time, and not only upon analysis
line 121: fibroblast culture – mark as headline
line 150: currently the text continues from method description to results in one line – please include a paragraph break
Fig4 legend – please refer to the patients either as father and daughter or as patient 1 and 2 in this figure, no mix
line 135: not mentioned which patient cells are used here (probably patient 1)
line 137: the graph in D represents MDA, but neither lipid peroxidation itself nor oxidative stress. MDA is used as a marker of lipid peroxidation, but can also arise through other pathways; so the two are not synonymous.
“Oxidative stress production” is an inappropriate term anyways, as oxidative stress means elevated steady-state levels of ROS, which can arise by higher ROS production or lower ROS scavenging.
line 152: were lower in the father compared to both healthy control and daughter
line 157: levels in gingival fibroblasts
line 161: you have not investigated ROS production, so you cannot make the claim that lipid peroxidation is caused by ROS overproduction – it’s very likely but not the only possibility
162: oxidative stress overproduction is a strange term – oxidative stress exists but is not produced or even overproduced
line 195ff: About laminin 5…before for HGF occurred – this sentence is grammatically a mess, please check
line 198: “Researchers agree that the connective tissue presents a high quantity of collagen” – I assume you mean the increase in connective tissue in this disease, not connective tissue in general.
line 202: not sure what you mean by “oxidative stress is traditionally related to apoptosis and cellular death” – cell death is one out of many results of oxidative stress, and I don’t see any reason to claim this “the traditional” one (especially since oxidative stress is not very old as a term anyways)
line 202f: this sounds like there’s only two effects of ROS: apoptosis or collagen production. There’s a little bit more discovered by now, so better rephrase
line 206: you have not measured antioxidant activity – the CoQ10 levels represent something else, and the typical cellular antioxidants such as Gluthation or SOD2 or catalase you have not assessed.
line 209: with two patients checked you cannot state that increased collagen production, higher MDA levels and lower CoQ10 are connected with more severe symptoms, especially since one of the two patients did not have reduced CoQ10 or increased MDA compared to controls.
line 216: since you have a patients showing symptoms without any oxidative stress signs, the use of antioxidants should not help this patient. And just increasing intracellular antioxidants could mess up a lot of intracellular signaling, ROS are not bad by default!
Author Response
Comments Reviewer 2
Thank you for your comments that improve our paper
Lourdes Roman-Malo and coworkers present in this manuscript two related patients suffering from gingival overgrowth. The study describes the patients’ symptoms and investigates several parameters related to the disease in cultured fibroblasts of the patients. The authors find excessive collagen production and indications of oxidative stress in the patient cells.
While from the results presented here it becomes clear that the authors investigate an interesting and potentially central molecular aspect of the disease, this manuscript is combining the clinical aspects (patient symptoms and treatment) with the in vitro investigation on cell level. Unfortunately both parts suffer, as neither is properly presented and discussed.
The manuscript text is often imprecise (although the general meaning becomes clear), and the statements exaggerated. A thorough checking of every sentence for its actual appropriateness would be recommendable, and also the clear separation of actual results and hypothetical interpretation.
I recommend to either concentrate on the description of patient pathology, also elaborating on the treatment details and how it affected the development of the disease – or on the molecular reasons for familial hereditary gingivitis, leaving only a brief description of the patients, but conducting a range of more experiments to address the molecular processes.
My suggestion for a minimal set of experiments for the latter:
- quantification of ROS production using any of the usual fluorescent measurements in wt and patient1/2 fibros
- quantification of mitochondrial activity to confirm the low CoQ10 levels (e.g. MTA test)
- quantification of oxidative defenses (e.g. SOD2 or Glutathione)
Thanks for these proposals but our objective was to study the histological alterations and the collagen synthesis in the cultured gingival fibroblasts and how it can be modified by an oxidant and an antioxidant.
Major critics:
- The description of the various types of HGF in the introduction is a bit confuse – after reading it I don’t understand which types exist and how they differ. Canges has been included
- CoQ10 is not an antioxidant used by the cell to defend against oxidative stress, but a part of the mitochondrial electron transport chain. It can be of course added as an antioxidant, but the quantification of endogenous CoQ10 does tell about the respiratory capacity, not about the antioxidant defense system of the cell. Please discuss (and preferably address experimentally) the mitochondrial etiology of the ROS stress and collagen production. Gingival overgrowth is a typical adverse effect of cyclosporin, that is known to screw up mitochondrial Calcium handling, again suggesting an alteration of mitochondrial function as the cause of the disease Our main goal was to study how the CoQ10 can modify in the fibroblasts culture the collagen production
- The patient’s treatment is elaborated in the discussion, but has actually not been the topic of the manuscript results. The paragraph has been deleted
- the fact that the patient’s fibroblasts produce excessive collagen also after culturing is not really discussed. As the cells are cultured in high glucose DMEM they are less dependent on mitochondrial oxidation, but still they suffer from oxidative stress and produce excessive collagen. That’s highly interesting and should not be ignored! High glucose DMEM is one of the most used medium and all the results were obtained in the same enviroment
- The daughter has symptoms and collagen levels are elevated in her fibros, but these cells do not exhibit a decrease in CoQ10 and also not any increase in MDA. Thus the oxidative stress appears to kick in only at later stages of the disease and cannot be the cause of it – unless the cultured cells do not represent well the state in the tissue.
Please discuss this thoroughly, as the central message (oxidative stress as a cause of HGF) is very much questioned by this patient’s results! The main differences between the father and the daughter are obviously the age and the degree of gingival overgrowth; this could explain the differences in these results. We include these aspects in the discussion
- Whether the inheritance is recessive cannot be clearly judged by the absence of the symptoms in a 10 or 12 year old child, if the disease starts usually with the first permanent teeth – potentially it starts to show a bit later than usual. The father noticed problems only at age 45, so potentially there is some variations in the onset and progression. Both daughters had just two years of differences in age, so we think that if the youngest has an absence of symptoms she will not develop a gingival overgrowth in the future.
small things: All the suggestions have been attended
line 2: here “gingival hereditary fibromatosis”, later on always “hereditary gingival fibromatosis”
line 3: in vitro and in vivo italic and without dash – all over the manuscript
line 30: should “natural” be rather “unnatural”?
line 32: show -> shows
line 33: issues -> problems, difficulties or similar
paragraph 29-40: this is somehow repetitive, twice mentioning the various degrees of overgrowth and the difficulties in speaking
line 45f: unclear whether Cowden, Jones, Goltz-Gorlin etc. is ONE syndrome or many syndromes? The whole sentence is a very long list that would be nicer spread over several sentences
line 48: “Also is associated”, something missing here, how about: “HGF is also associated”. In the same sentence “ingestion of drugs” probably means
line 49: the differential diagnosis -> a differential diagnosis
line 52: oral manifestations of the idiopathic type – which idiopathic type? Do you refer here to the autosomal dominant one? And are not all manifestations oral?
line 55: “inflammation, “ comma needed
paragraph 56ff: unclear to me whether the described lesions are belonging to the idiopathic type – if so, the topic suddenly jumps to familial cases in the next sentence – are they also idiopathic? If they refer to some other type please clarify which.
line 60: “that is formed”
line 68: this condition is related to a mechanism- how is it related?
line 70: the cell phenotype -> their phenotype
line 71: “and respond to the local stress depending on the metabolic conditions” – what exactly would you like to say here? Which local stress? Which metabolic conditions? How do the fibros react?
line 72: oxidative stress does not reflect the balance of ROS and antioxidants, but is an imbalance of these two
line 73: disturbances in the normal redox state could be also reduced ROS that are absolutely not harming any molecules, but only mess up the signaling
line 78: “we try to” – I assume you have succeeded with your try?!
line 83: does the informed consent needs to be mentioned here? it’s one case and not several cases The daughter is under-age and according to Spanish legislation we need to have the fathers informed consent
line 84f: no other complaints such as plain, bleeding or halitosis were present
line 85: He -> The patient… begun 4-5 years ago and progressed slowly
line 86: and he did not take any medication
line 87: within normal limits
line 99: severe enlargement of what?
line 105: “Histology” is a bit lonely – is it supposed to be a headline?
line 108: order information of the antibody missing
line 110: the epithelium was wider than what?
line 111: some characteristics of EMT
line 112ff: a bit weirdly phrased: the patients had the lack of laminin 5 all the time, and not only upon analysis
line 121: fibroblast culture – mark as headline
line 150: currently the text continues from method description to results in one line – please include a paragraph break
Fig4 legend – please refer to the patients either as father and daughter or as patient 1 and 2 in this figure, no mix
line 135: not mentioned which patient cells are used here (probably patient 1)
line 137: the graph in D represents MDA, but neither lipid peroxidation itself nor oxidative stress. MDA is used as a marker of lipid peroxidation, but can also arise through other pathways; so the two are not synonymous.
“Oxidative stress production” is an inappropriate term anyways, as oxidative stress means elevated steady-state levels of ROS, which can arise by higher ROS production or lower ROS scavenging.
line 152: were lower in the father compared to both healthy control and daughter
line 157: levels in gingival fibroblasts
line 161: you have not investigated ROS production, so you cannot make the claim that lipid peroxidation is caused by ROS overproduction – it’s very likely but not the only possibility
162: oxidative stress overproduction is a strange term – oxidative stress exists but is not produced or even overproduced
line 195ff: About laminin 5…before for HGF occurred – this sentence is grammatically a mess, please check
line 198: “Researchers agree that the connective tissue presents a high quantity of collagen” – I assume you mean the increase in connective tissue in this disease, not connective tissue in general.
line 202: not sure what you mean by “oxidative stress is traditionally related to apoptosis and cellular death” – cell death is one out of many results of oxidative stress, and I don’t see any reason to claim this “the traditional” one (especially since oxidative stress is not very old as a term anyways)
line 202f: this sounds like there’s only two effects of ROS: apoptosis or collagen production. There’s a little bit more discovered by now, so better rephrase
line 206: you have not measured antioxidant activity – the CoQ10 levels represent something else, and the typical cellular antioxidants such as Gluthation or SOD2 or catalase you have not assessed.
line 209: with two patients checked you cannot state that increased collagen production, higher MDA levels and lower CoQ10 are connected with more severe symptoms, especially since one of the two patients did not have reduced CoQ10 or increased MDA compared to controls.
line 216: since you have a patients showing symptoms without any oxidative stress signs, the use of antioxidants should not help this patient. And just increasing intracellular antioxidants could mess up a lot of intracellular signaling, ROS are not bad by default
Round 2
Reviewer 2 Report
Dear authors,
this manuscript is significantly improved now: the introduction is much clearer, the results are described more conclusive and the discussion is limited to teh rpesented results and more consistent. Still, I'd recommend a language check for this manuscript as it contains some rather innovative grammar.